# A Sampling-Based Unfixed Orientation Search Method for Dual Manipulator Cooperative Manufacturing

**DOI:** 10.3390/s22072502

**Published:** 2022-03-24

**Authors:** Chang Su, Jianfeng Xu

**Affiliations:** 1School of Mechanical Science & Engineering, Huazhong University of Science & Technology, Wuhan 430074, China; 2State Key Laboratory of Digital Manufacturing Equipment and Technology, Huazhong University of Science & Technology, Wuhan 430074, China; jfxu@hust.edu.cn

**Keywords:** dual manipulator system, cooperative manufacturing, collision-free path planning, non-linear optimization programming, minimum distance prediction

## Abstract

The case of dual manipulators with shared workspace, asynchronous manufacturing tasks, and independent objects is named a dual manipulator cooperative manufacturing system, which requires collision-free path planning as a vital issue in terms of safety and efficiency. This paper combines the mathematical modeling method with the time sampling method in the classification of robot path-planning algorithms. Through this attempt we can achieve an optimal local search path during each sampling period interval. Our strategy is to build the corresponding non-linear optimization functions set based on the motion characteristics of the dual manipulator system. In this way, the path-planning problem can be turned into a purely mathematical problem of solving the non-linear optimization programming equations set. The spatial geometric analysis is used to linearize the predicted dual-manipulator minimum distance equation, thus linearizing the non-linear optimization equations set. Finally, this system of linear optimization equations will be mapped directly into a virtual Euclidean space and then solved intuitively using the spatial geometry theory. By simulation and comparing with the previous strategies, we find that the planning results of the newly proposed planning strategy are smoother and have shorter deviations as well as a higher algorithmic efficiency in terms of spatial geometric properties.

## 1. Introduction

### 1.1. Subsection

At present, six-degree-of-freedom (6-DOF) industrial manipulators play an increasingly important role in automated manufacturing due to numerous advantages, including the provision of tireless repetitive labor, faster-moving speeds, and a higher accuracy performance [1]. Thus, tasks requiring numerous workers can undoubtedly be accomplished cooperatively by multiple manipulators, meaning that multi-manipulators will not only work side-by-side but as dyads and teams. Analogous to the definition of Human–Robot Interaction (HRI) [2,3], the case of a dual manipulator manufacturing system can be divided into cooperation and collaboration according to the arrangement of the tasks. For the case of a collaborative manufacturing system, all manipulators together with the executed mechanism constitute a complete multi-degree-of-freedom closed-loop or parallel manufacturing system. Thus, the non-collision manufacturing control strategy under such circumstances can be transformed into the currently mature internal obstacle avoidance strategy [4]. However, for the case of the multiple manipulator cooperative manufacturing system, problems and limitations arise in the small overlapping workspaces that accommodate numerous cooperative manipulators, resulting in collisions if no related countermeasures are put in place. Under this circumstance, a path-planning algorithm that can circumvent the constraints of the mechanical structure, simplify the planning space, and determine the paths of all manipulators involved in the cooperative system in real-time will be the kernel.

### 1.2. Relative Work

A 6-DOF industrial manipulator implies a type of serial chain robot with six revolving joints, and its structure is much more complex than that of a mobile robot. Since a single manipulator can be modeled as a particle in its own configuration space (C-space), the particle-based path-planning methods [5] can be used for single manipulator path-planning directly. Limited progress has been made for the non-collision path planning of multiple cooperating manipulators in C-space. Obstacles were modeled as simple geometries by Jia et al. [6], which will then be projected into the C-space of a manipulator. This method is not effective for multiple manipulators due to their complex structures. Following on from this, Yu et al. [7] divided the C-space into multiple layers based on specific dimensions and implemented a rapidly created C-space grid map on each layer. However, because of the massive amount of elements in the maps, the whole algorithm tends to be time-consuming. Li et al. [8] planned collision-free paths for 2D horizontally articulated dual-arm robots by dividing the C-space of each robot into multiple blocks, marking the obstacle space and free space at different time intervals, and mapping the search method to seek out an optimal path in free space. The method was too computationally expensive for real-time applications, and it is difficult to project a 6-DOF manipulator body into the C-space of another 6-DOF manipulator. Harada et al. [9] proposed a general manipulation planner for a dual-arm industrial manipulator, which combined the C-space of the arms and obstacle space into one overall search map. The method mainly focused on each arm’s trajectory arrangement and movement order to allow a target object to pass from starting point to endpoint. However, the possibility of collision between the two working arms was not considered.

Hence, how to explore a simple, effective, and reliable execution space path-planning algorithm has become one of the more and more enthusiastic focuses in recent decades. An online collision-free trajectory generation algorithm for dual-arm robots was established by Lee et al. [10] by setting up a Virtual Road Map (VRM) in the execution space and refreshing the map with a new collision-free path. However, the execution time is extended due to difficulties in setting up the VRM, and raising the method to multiple robots is challenging. Larsen et al. [11] divided the working area into different zones and set up obstacle models within each zone. A master–slave method was then used to choose a free path; however, dynamic environments cannot be accommodated since the obstacle must be static. Cohen et al. [12] combined the C-space with execution space to build a “motion space”. They used the Lazy Weighted A* method to plan a non-collision path in an environment containing N-manipulators. Still, only the path planning for one robot is allowed at a time, and the planning process is performed offline. While the ARA* method was also tested to search for the optimal path in the workspace (execution space) of a manipulator, the study focused only on dual arms performing the same task. It did not consider the coordination of different tasks.

Additional methods have been proposed for the non-collision path planning of multiple manipulators. Chiddarwar et al. [13] used the A* method to plan a collision-free path in C-space without considering the motion of other robots, and a Path Modification Sequence (PMS) method was then proposed to arrange the moving sequence of each robot, resulting in a much longer execution time. Another method proposed by Afaghani et al. [14] used a collision map to detect the collisions between two robots to avoid deadlocks, which can occur if one robot becomes an obstacle to another. Unfortunately, the method simply delays the movement of the robot to avoid collision. Rodríguez et al. [15] suggested an approach based on a variation of the Probabilistic Road Map, called the Probabilistic Road Map with Obstacles (PRMwO). The method does not exclude collision samples with removable objects, but instead classifies them as collided obstacle(s) and allows the search for accessible paths highlighting which objects must be removed from the workspace to make a valid path. While this approach removes any obstacles along the path of the working manipulator, it does not enable the manipulator to actually bypass obstacles. Habibnejad et al. [16] used the Artificial Potential Field (APF) method to plan the paths of multiple cooperative manipulators on mobile bases. However, the study focused solely on the path of the end effector (EEF) and did not consider the entire arm, thus lacking important systemic considerations.

A preliminary conclusion can be drawn from the above review that when it comes to the path planning of dual or multiple industrial manipulators, the negative influence, due to the above characteristics such as model complexity, high dimensionality, and complex planning space, will become even more critical. Multi-manipulator path-planning algorithms can be instantly divided into a map search-based method, time-sampling-based method, mathematical model method, and others [17]. The map search-based method [18,19,20] is a global optimization algorithm that will cost a colossal amount of time in planning map construction. In comparison, the time-sampling-based method [21,22,23] divides the whole planning period into several orderly but isolated intervals and splices all the calculated outcomes of each interval into a complete result path or planning map in order.

## 2. Previous Work and Paper Organization

In our previous work, a novel method called the Sampling-based Operation Space Map Search (SbOSMS) method (picture c in Figure 1), which combines the map search method (picture a in Figure 1) with the time-sampling-based method (picture b in Figure 1), was proposed. Equivalently, a local planning map was established during each time interval, and an optimal local path was determined following the local planning map. All chronologically planned path segments are glued together to generate a collision-free path, which is highly effective at the cost of forgoing the global optimum.

When focusing on how to build a suitable search map during each sampling period, we find that the nodes constituting the search map are several specific points selected by a particular rule on the surface of the entire search space (usually a sphere space) in that sampling period (Figure 2). Therefore, calculating the optimal path points in the whole search space (sphere space) surface rather than merely the constructed search map will be the central focus of this paper’s study, as shown in picture d in Figure 1.

In this paper, a brief path-planning method review of the dual manipulator cooperative system is shown in Section 1. Section 2 provides a concise snapshot of our initial efforts on this issue and the limitations we encountered. Section 3 proposes a mathematical model for the non-linear optimization of dual manipulator path planning in detail. To convert all equation terms in the set of non-linear optimization equations into equations with the same independent variables in Section 4, we use spatial geometry to linearize the minimum distance prediction equation within a specific sampling period. Subsequently, in Section 5, spatial vector geometry is used to quickly solve the set of equations with the same independent variables that have been transformed in Section 4. Section 6 presents the simulation results and relevant analyses. Finally, in Section 7, some conclusions are offered.

## 3. Mathematic Model

Here, the following motion for an arbitrary dual manipulator cooperative system is preset: manipulator A starts from $P_{A}^{start}$ heading towards $P_{A}^{target}$, in parallel, manipulator arm B is also moving on its initial path from the starting point $P_{B}^{start}$ to the target point $P_{B}^{target}$. As we consider generating reasonable geometrical paths for all manipulators, the processing of finding the optimal position points around the current positions in their Cartesian space with a distance of $L_{s}$ will be focused on during each planning period. Figure 3 visualizes the core for building the mathematical model of our proposal with the circumvention strategy of the dual manipulator system in a certain sampling period $t\sim \left(t+t_{s}\right)$. Assume the search step length during a sampling time interval $t_{s}$ in the planning space for both manipulators is $L_{s}^{t}$ and we make it constant during every sampling period before reaching the target $L_{s}^{t}=L_{s}\;\left(t\in \left[1,2,\ldots ,N\right]\right)$.

When the minimum distance between the two manipulators is smaller than a predefined value, the dual manipulators are considered in the danger area in which there is a high probability of collision. The overlap between the two working areas is defined as the danger area. Therefore, when the two manipulators move into the danger area, the EEFs of the dual manipulators should move towards their respective bases to avoid the possible collision. Under this circumstance, the two manipulators are regarded as obstacles to each other. According to Figure 4, the non-linear optimization equations set of dual manipulator path planning will be established:
(1)$$\begin{array}{cc} min & \underset{i\in (A,B)}{\sum }\mu_{i}\left[\left(\alpha_{i}V_{ie}^{t}+\beta_{i}\left(P_{i}^{base}-P_{ie}^{t}\right)\right)\cdot {\left(P_{ie}^{t}-P_{ie}^{t+t_{s}}\right)}^{T}+{∥P_{i}^{target}-P_{ie}^{t+t_{s}}∥}^{2}\right]\\ s.t. & \begin{array}{c} P_{Ae}^{t}-P_{Ae}^{t+t_{s}}=steplength;\\ P_{Be}^{t}-P_{Be}^{t+t_{s}}=steplength;\\ d_{t+t_{s}}^{min}\geq L_{min} \end{array} \end{array}$$
where $V_{Ae}^{t}$, $V_{Be}^{t}$ are the linear velocities of EEFs of manipulator *A* and *B* at time *t*; 

$P_{Ae}^{t}$, $P_{Be}^{t}$ are the Cartesian coordinates of EEFs of manipulator *A* and *B* at time *t*; 

$P_{Ae}^{t+t_{s}}$, $P_{Be}^{t+t_{s}}$ are the Cartesian coordinates of EEFs of manipulator *A* and *B* at time ($t+t_{s}$); 

$P_{A}^{target}$, $P_{B}^{target}$ are the target Cartesian coordinates of manipulator *A* and *B*; 

steplength indicates the length of the search step of the manipulators’ EEF within one sampling cycle $t_{s}$; 

$d_{t+t_{s}}^{min}\left(i\in int\left(1\sim n\right)\right)$ is the minimum distance between the two manipulators during sampling period $t\sim \left(t+t_{s}\right)$; 

$L_{min}$ is the threshold of the minimum distance value;

$\mu_{A}$, $\mu_{B}$ are the positive circumventing weighting factors that meet $\mu_{A}+\mu_{B}=1$. $\mu_{A}=0$ means manipulator A will always follows its initial path without modification and vice versa.

$\alpha_{A}$, $\alpha_{B}$, $\beta_{A}$ and $\beta_{B}$ are positive deviation direction weighting coefficients for manipulators A and B, which take values in the set of {0,1}. 

We simplify the Formula (1):$$\begin{array}{cc} min\underset{i\in \left(A,B\right)}{\sum }\mu_{i} & \left[\left(\alpha_{i}V_{ie}^{t}+\beta_{i}\left(P_{i}^{base}-P_{ie}^{t}\right)\right)\cdot {\left(P_{ie}^{t}-P_{ie}^{t+t_{s}}\right)}^{T}+∥P_{i}^{target}-P_{ie}^{t+t_{s}}∥{}^{2}\right]\\ & ∼min\underset{i\in \left(A,B\right)}{\sum }\mu_{i}\left[\left(\alpha_{i}V_{ie}^{t}+\beta_{i}\left(P_{i}^{base}-P_{ie}^{t}\right)\right)\cdot {\left(P_{ie}^{t}-P_{ie}^{t+t_{s}}\right)}^{T}+\left(P_{i}^{target}-P_{ie}^{t}\right){\left(P_{ie}^{t}-P_{ie}^{t+t_{s}}\right)}^{T}\right]\\ & ∼min\underset{i\in \left(A,B\right)}{\sum }\mu_{i}\left[\frac{\alpha_{i}V_{ie}^{t}+\beta_{i}\left(P_{i}^{base}-P_{ie}^{t}\right)}{∥\alpha_{i}V_{ie}^{t}+\beta_{i}\left(P_{i}^{base}-P_{ie}^{t}\right)∥}\cdot {\left(P_{ie}^{t}-P_{ie}^{t+t_{s}}\right)}^{T}+\left(P_{i}^{target}-P_{ie}^{t}\right){\left(P_{ie}^{t}-P_{ie}^{t+t_{s}}\right)}^{T}\right]\\ & ∼\;min\underset{i\in \left(A,B\right)}{\sum }\mu_{i}\left[\frac{\alpha_{i}V_{ie}^{t}+\beta_{i}\left(P_{i}^{base}-P_{ie}^{t}\right)}{∥\alpha_{i}V_{ie}^{t}+\beta_{i}\left(P_{i}^{base}-P_{ie}^{t}\right)∥}+\left(P_{i}^{target}-P_{ie}^{t}\right)\right]\cdot {\left(\frac{P_{ie}^{t}-P_{ie}^{t+t_{s}}}{steplength}\right)}^{T} \end{array}$$

Here, we can set
$$a_{1}=\mu_{A}\left[\frac{\alpha_{A}V_{Ae}^{t}+\beta_{A}\left(P_{A}^{base}-P_{Ae}^{t}\right)}{∥\alpha_{A}V_{Ae}^{t}+\beta_{A}\left(P_{A}^{base}-P_{Ae}^{t}\right)∥}+\left(P_{A}^{target}-P_{Ae}^{t}\right)\right];\;a_{2}=\mu_{B}\left[\frac{\alpha_{B}V_{Be}^{t}+\beta_{B}\left(P_{B}^{base}-P_{Be}^{t}\right)}{∥\alpha_{B}V_{Be}^{t}+\beta_{B}\left(P_{B}^{base}-P_{Be}^{t}\right)∥}+\left(P_{B}^{target}-P_{Be}^{t}\right)\right];\;x_{1}=\frac{P_{Ae}^{t}-P_{Ae}^{t+t_{s}}}{steplength};x_{2}=\frac{P_{Be}^{t}-P_{Be}^{t+t_{s}}}{steplength};$$

The objective function in the initial mathematical model (1) can be simplified as:(2)$$\begin{array}{cc} min & a_{1}\cdot x_{1}^{T}+a_{2}\cdot x_{2}^{T} \end{array}$$

The restricted condition functions in the initial mathematical model (1) can be translated into:(3)$$\begin{array}{cc} s.t. & \begin{array}{c} \begin{array}{c} ∥x_{1}∥=1\\ ∥x_{2}∥=1 \end{array}\\ d_{t+t_{s}}^{min}\geq L_{min} \end{array} \end{array}$$

For Formulas (2) and (3), if we can find a method to convert the inequality expressions of $d_{t+t_{s}}^{min}\geq L_{min}$ into the inequality equation with $\left(x_{1},x_{2}\right)$ as a variable, the entire initial mathematic model can be converted into a vector optimization model with corresponding variables $\left(x_{1},x_{2}\right)$. Under this case, we can solve the vector optimization model using existing mature mathematical tools efficiently.

Next, we will describe in detail the specific methods for determining the values of $\alpha_{A}$, $\alpha_{B}$, $\beta_{A}$ and $\beta_{B}$. Back to Figure 4, there are three direct external factors that affect the obstacle avoidance strategy of an industrial manipulator: the current velocity of motion $V_{ie}^{t}$, the target point $P_{i}^{target}$ and the base of the manipulator $P_{i}^{base}$. In Formula (1), $\left[\begin{array}{cc} min & \alpha_{i}V_{ie}^{t}\cdot \end{array}{\left(P_{ie}^{t}-P_{ie}^{t+t_{s}}\right)}^{T}\right]$ guarantees to maintain the motion inertia of the manipulator as much as possible during the current planning sampling period, $\left[min{∥P_{i}^{target}-P_{ie}^{t+t_{s}}∥}^{2}\right]$ drives the manipulator towards its target point along the feasible shortest path, and $\left[\begin{array}{cc} min & \left(\beta_{i}\left(P_{i}^{base}-P_{ie}^{t}\right)\right)\cdot {\left(P_{ie}^{t}-P_{ie}^{t+t_{s}}\right)}^{T} \end{array}\right]$ gives the manipulator a tendency to go to its safest place—base. Take manipulator A for example, the geometric relationship between the three spatial vectors ($P_{Be}^{t}-P_{Ae}^{t}$), ($P_{A}^{target}-P_{Ae}^{t}$) and $\left(P_{A}^{base}-P_{Ae}^{t}\right)$ can be used to determine the values of two variables $\alpha_{A}$ and $\beta_{A}$.
(4)$$P_{A}^{target}-P_{Ae}^{t}=\omega *\left(P_{Be}^{t}-P_{Ae}^{t}\right)+\varphi *\left(P_{A}^{base}-P_{Ae}^{t}\right)+\tau *\left(\left(P_{Be}^{t}-P_{Ae}^{t}\right)\times \left(P_{A}^{base}-P_{Ae}^{t}\right)\right)$$

If ω>0 and φ>0, ($P_{A}^{target}-P_{Ae}^{t}$) is in the region of space defined by vectors $\left(P_{A}^{base}-P_{Ae}^{t}\right)$ and $\left(P_{Be}^{t}-P_{Ae}^{t}\right)$. In this case, we can define $\alpha_{A}=1$, $\beta_{A}=0$, which means the manipulator A can circumvent the vector $\left(P_{Be}^{t}-P_{Ae}^{t}\right)$, a vector representing the dangerous spatial positions of the current dual manipulator system, by simply following the current motion tendency.

If ω<0 || φ<0, ($P_{A}^{target}-P_{Ae}^{t}$) is out of the space region defined by vectors $\left(P_{A}^{base}-P_{Ae}^{t}\right)$ and $\left(P_{Be}^{t}-P_{Ae}^{t}\right)$. In this case, the mere maintenance of the motion trend towards the termination points do not guarantee manipulator A circumventing vector $\left(P_{Be}^{t}-P_{Ae}^{t}\right)$. Therefore, forgoing the motion option of heading towards the terminals and temporarily seeking avoidance towards the base of the manipulator A is a better motion strategy, making $\alpha_{A}=0$, $\beta_{A}=1$. 

Such derivations and conclusions also apply to manipulator *B*

## 4. Minimum Distance Prediction

Collision detection is a fundamental theoretical research area of robotics, focusing on establishing a reasonable manipulator model and calculating the minimum distance between the manipulator and the obstacle. Collision detection can generally be divided into static collision detection, dynamic collision detection, and continuous collision detection [24]. Static collision detection is for the case where the manipulator and the obstacle are in a static state. In this case, collision detection can be modeled as the process of computing the minimum distance between two static geometric models in Cartesian space. Dynamic collision detection is where at least one of the manipulators and the obstacle is moving. The dynamic collision detection problem can be transformed into static collision detection because continuous time can be discretized. Both the manipulator and the obstacle can be considered static at each time interval. Continuous collision detection uses time as a variable to study the trend of the minimum distance between the manipulator and the obstacle in a given time interval.

Obviously, by observing Formulas (2) and (3), the optimal function and condition functions in the optimization equations are all contain the same variable term $\left(x_{1},x_{2}\right)$. However, the inequality function $d_{t+t_{s}}^{min}\geq L_{min}$ cannot connect with the optimization function via the common variables through the existing discrete collision detection algorithm. The static collision detection and dynamic collision detection methods are not satisfied to determine the relationship between the minimum distance and the movement trend of path-planned manipulators. Therefore, we need to find a continuous collision detection algorithm that transforms the constraint functions of the minimum distance into inequality functions that take $\left(x_{1},x_{2}\right)$ as variables, so that the whole set of optimization equations can be transformed into a function set with the same vector variables, which will be easy for us to solve.

The first step for collision detection is robot modeling. For multi-axis chain manipulators, there are many mature modeling methods [25]. Here, we use the Sweep Sphere Volume (SSV) [26] method, which envelops each link of a manipulator’s arm with a capsule body.

For $d_{t+t_{s}}^{min}\geq L_{min}$, it is equivalent to:(5)$$d_{t+t_{s}}^{min}\geq L_{min}\Leftrightarrow d_{t}^{min}-\Delta d_{t+t_{s}}\geq L_{min}\Leftrightarrow \Delta d_{t+t_{s}}\leq d_{t}^{min}-L_{min}$$
where $d_{t}^{min}$ is the shortest distance between manipulator and obstacle(s) at current time instant t, so its value must be known, and $L_{min}$ is a predefined value. Thus, we need to figure out how to express $\Delta d_{t+t_{s}}$ in some kind terms of $\left(x_{1},x_{2}\right)$ in the dual manipulator situation. In the following content of this section, we will propose a novel minimum distance prediction (MDP) method to meet our requirements.

MDP method is derived from the algebraic geometry (AG) method of collision detection. Via the AG method, the problem of calculating the minimum distance between objects is transformed into the calculation of distance between abstract space geometries. From Figure 5, we can observe that the movements of the minimum-distance points of the two manipulators along their respective mechanical arms are continuous, and there exists only one minimum-distance point for each manipulator at a time. At the same time, the calculation of minimum distance between two manipulators can be simplified into the calculation of minimum distance between two space segments according to the principle of collision detection. Therefore, based on the knowledge of the positions and the velocities of the two minimum-distance points under the current instant, we can predict the possible locations of the minimum-distance points and the distance between them in the next scanning period. Furthermore, the velocities of two manipulator EEFs and the positions of mechanical arms on which the two minimum-distance points will stand under the present moment can determine the variation of the minimum distance during the next scanning period.

As shown in Figure 5, according to the current moving state of the two minimum-distance points of the two manipulators, we can get:(6)$$\Delta d_{t+t_{s}}=V_{BA}^{t+t_{s}}\cdot \frac{P_{BA}^{t}{}^{T}}{∥P_{BA}^{t}∥}\cdot t_{s}=\left(V_{A}^{t+t_{s}}-V_{B}^{t+t_{s}}\right)\cdot \frac{P_{BA}^{t}{}^{T}}{∥P_{BA}^{t}∥}\cdot t_{s}$$
where $P_{BA}^{t}=P_{Bc}^{t}-P_{Ac}^{t}$, $P_{Ac}^{t}$ and $P_{Bc}^{t}$ represent the minimum distance point position vectors of manipulator A and B under the global coordinate system. According to the different types of manipulators, the relationships between $\left(V_{A}^{t+t_{s}}\;\&\;V_{B}^{t+t_{s}}\right)$ and $\left(x_{1},x_{2}\right)$ are different, while the points $P_{Ac}^{t}$ and $P_{Bc}^{t}$ will change along with the manipulator body links with time.

Meanwhile, the relationship between the velocities of minimum-distance points and those of manipulator EEFs can be obtained referring to:(7)$$\begin{array}{c} {\left(V_{A}^{t+t_{s}}\right)}^{T}=\left(J_{A}J_{Ae}^{-1}\right)\cdot {\left(V_{Ae}^{t+t_{s}}\right)}^{T}\\ {\left(V_{B}^{t+t_{s}}\right)}^{T}=\left(J_{B}J_{Be}^{-1}\right)\cdot {\left(V_{Be}^{t+t_{s}}\right)}^{T} \end{array}$$
where $V_{Ac}^{t+t_{s}}$ and $V_{Bc}^{t+t_{s}}$ are the velocities of $P_{Ac}^{t}$ and $P_{Bc}^{t}$; $V_{Ae}^{t+t_{s}}$ and $V_{Be}^{t+t_{s}}$ are the velocities of the EEFs of manipulator A and manipulator B, individually; $P_{A}^{t}$ is the coordinate of $P_{Ac}^{t}$ and $P_{B}^{t}$ is the coordinate of $P_{Bc}^{t}$.$$\begin{array}{cc} \Delta d_{t+1}=\frac{P_{BA}^{t}}{∥P_{BA}^{t}∥}\cdot & {\left(V_{A}^{t+t_{s}}-V_{B}^{t+t_{s}}\right)}^{T}\cdot \Delta t\\ & =\frac{P_{BA}^{t}}{∥P_{BA}^{t}∥}\cdot \left[\left(J_{A}J_{Ae}^{-1}\right)\cdot {\left(V_{Ae}^{t+t_{s}}\right)}^{T}-\left(J_{B}J_{Be}^{-1}\right)\cdot {\left(V_{Be}^{t+t_{s}}\right)}^{T}\right]\cdot \Delta t\\ & =\left[\frac{P_{BA}^{t}}{∥P_{BA}^{t}∥}\cdot \left(J_{A}J_{Ae}^{-1}\right)\right]\cdot \left[{\left(V_{Ae}^{t+t_{s}}\right)}^{T}\cdot \Delta t\right]-\left[\frac{P_{BA}^{t}}{∥P_{BA}^{t}∥}\cdot \left(J_{B}J_{Be}^{-1}\right)\right]\\ & \cdot \left[{\left(V_{Be}^{t+t_{s}}\right)}^{T}\cdot \Delta t\right]\\ & =\left[\frac{P_{BA}^{t}}{∥P_{BA}^{t}∥}\cdot \left(J_{A}J_{Ae}^{-1}\right)\right]\cdot {\left(P_{Ae}^{t+t_{s}}-P_{Ae}^{t}\right)}^{T}-\left[\frac{P_{BA}^{t}}{∥P_{BA}^{t}∥}\cdot \left(J_{B}J_{Be}^{-1}\right)\right]\\ & \cdot {\left(P_{Be}^{t+t_{s}}-P_{Be}^{t}\right)}^{T}\\ & =\left[\frac{P_{BA}^{t}}{∥P_{BA}^{t}∥}\cdot \left(J_{B}J_{Be}^{-1}\right)\right]\cdot \left(x_{1}^{T}\cdot steplength\right)+\left[\frac{P_{AB}^{t}}{∥P_{BA}^{t}∥}\cdot \left(J_{A}J_{Ae}^{-1}\right)\right]\\ & \cdot \left(x_{2}^{T}\cdot steplength\right)\leq d_{t}^{min}-L_{min} \end{array}$$

Thus:(8)$$\left[\frac{P_{BA}^{t}}{∥P_{BA}^{t}∥}\cdot \left(J_{B}J_{Be}^{-1}\right)\right]\cdot x_{2}^{T}+\left[\frac{P_{AB}^{t}}{∥P_{AB}^{t}∥}\cdot \left(J_{A}J_{Ae}^{-1}\right)\right]\cdot x_{1}^{T}\leq \frac{d_{t}^{min}-L_{min}}{steplength}$$

Set:$$b_{1}=\frac{P_{AB}^{t}}{∥P_{AB}^{t}∥}\cdot \left(J_{A}J_{Ae}^{-1}\right);b_{2}=\frac{P_{BA}^{t}}{∥P_{BA}^{t}∥}\cdot \left(J_{B}J_{Be}^{-1}\right);D=\frac{d_{t}^{min}-L_{min}}{steplength};$$

The initial mathematical model of Formula (1) can be simplified as:(9)$$\begin{array}{cc} min & a_{1}\cdot x_{1}^{T}+a_{2}\cdot x_{2}^{T}\\ s.t. & \begin{array}{c} b_{1}\cdot x_{1}^{T}+b_{2}\cdot x_{2}^{T}\leq D\\ ∥x_{1}∥=1;∥x_{2}∥=1; \end{array} \end{array}$$

Obviously, it is a vector linear programming problem.

## 5. Sampling-Based Unfixed Orientation Search Method

This chapter will describe converting vector optimization mathematic models into a scalar optimization problem. For Formula (9), assume that $b_{1}\cdot x_{1}^{T}=x$, $b_{2}\cdot x_{2}^{T}=y$. Then, we try to transfer the target function into the function of the two independent variables, x and y. Now we take $b_{1}\cdot x_{1}^{T}=x$ as an example, the following operational formulas can be obtained based on the vector diagram shown in Figure 6:
$$c_{1}=\frac{x}{{∥b_{1}∥}^{2}}\cdot b_{1};\omega_{1}=\left(b_{1}\times a_{1}\right)\times b_{1};\;{\left(d_{1}\right)}_{max}=\sqrt{1-{\left(\frac{x}{∥b_{1}∥}\right)}^{2}}\cdot \frac{\omega_{1}}{∥\omega_{1}∥}\;{\left(d_{1}\right)}_{min}=-\sqrt{1-{\left(\frac{x}{∥b_{1}∥}\right)}^{2}}\cdot \frac{\omega_{1}}{∥\omega_{1}∥}\;\min\left(a_{1}\cdot x_{1}^{T}\right)=a_{1}\cdot \left(c_{1}+{\left(d_{1}\right)}_{min}\right)=a_{1}\cdot \left(\frac{x}{{∥b_{1}∥}^{2}}\cdot b_{1}-\sqrt{1-{\left(\frac{x}{∥b_{1}∥}\right)}^{2}}\cdot \frac{\omega_{1}}{∥\omega_{1}∥}\right)\;=\frac{a_{1}\cdot b_{1}^{T}}{{∥b_{1}∥}^{2}}x-\frac{a_{1}\cdot \omega_{1}^{T}}{∥b_{1}∥∥\omega_{1}∥}\sqrt{{∥b_{1}∥}^{2}-x^{2}}=\alpha_{1}x-\beta_{1}\sqrt{\gamma_{1}{}^{2}-x^{2}}$$
where:$$\alpha_{1}=\frac{a_{1}\cdot b_{1}^{T}}{{∥b_{1}∥}^{2}};\beta_{1}=\frac{a_{1}\cdot \omega_{1}^{T}}{∥b_{1}∥∥\omega_{1}∥}>0;\gamma_{1}=∥b_{1}∥>0\;\alpha_{2}=\frac{a_{2}\cdot b_{2}^{T}}{{∥b_{2}∥}^{2}};\beta_{2}=\frac{a_{2}\cdot \omega_{2}^{T}}{∥b_{1}∥∥\omega_{1}∥}>0;\;\gamma_{2}=∥b_{2}∥>0$$

Therefore, the initial mathematical model will eventually be transferred into the following form:(10)$$\begin{array}{cc} min & \alpha_{1}x-\beta_{1}\sqrt{\gamma_{1}{}^{2}-x^{2}}+\alpha_{2}y-\beta_{2}\sqrt{\gamma_{2}{}^{2}-y^{2}}\\ sub\;to & x+y\leq D \end{array}$$
where:$$a_{1}=\mu_{A}\left[\frac{\alpha_{A}V_{Ae}^{t}+\beta_{A}\left(P_{A}^{base}-P_{Ae}^{t}\right)}{∥\alpha_{A}V_{Ae}^{t}+\beta_{A}\left(P_{A}^{base}-P_{Ae}^{t}\right)∥}+\left(P_{A}^{target}-P_{Ae}^{t}\right)\right];\;a_{2}=\mu_{B}\left[\frac{\alpha_{B}V_{Be}^{t}+\beta_{B}\left(P_{B}^{base}-P_{Be}^{t}\right)}{∥\alpha_{B}V_{Be}^{t}+\beta_{B}\left(P_{B}^{base}-P_{Be}^{t}\right)∥}+\left(P_{B}^{target}-P_{Be}^{t}\right)\right];\;D=\frac{d_{t}^{min}-L_{min}}{steplength};\;x_{1}=\frac{P_{Ae}^{t}-P_{Ae}^{t+1}}{steplength};x_{2}=\frac{P_{Be}^{t}-P_{Be}^{t+1}}{steplength};\;b_{1}=\frac{P_{AB}^{t}}{∥P_{AB}^{t}∥}\cdot \left(J_{A}J_{Ae}^{-1}\right);b_{2}=\frac{P_{BA}^{t}}{∥P_{BA}^{t}∥}\cdot \left(J_{B}J_{Be}^{-1}\right);\;\;\alpha_{1}=\frac{a_{1}\cdot b_{1}^{T}}{{∥b_{1}∥}^{2}};\beta_{1}=\frac{a_{1}\cdot \omega_{1}^{T}}{∥b_{1}∥∥\omega_{1}∥};\;\gamma_{1}=∥b_{1}∥;\;\alpha_{2}=\frac{a_{2}\cdot b_{2}^{T}}{{∥b_{1}∥}^{2}};\beta_{2}=\frac{a_{2}\cdot \omega_{2}^{T}}{∥b_{2}∥∥\omega_{2}∥};\;\gamma_{2}=∥b_{2}∥;\;x=b_{1}\cdot x_{1}^{T};\;y=b_{2}\cdot x_{2}^{T}$$

Apparently, Formula (10) is a binary non-linear optimal solution equation that can be easily solved using “nlopt” open-source software library in the vs. programming platform.

## 6. Simulation and Analysis

In this paper, assuming 6-DOF industrial manipulators have standard D-H parameters, the simulation results for the dual manipulator systems will be analyzed individually. All the algorithms were programmed in C++ and run on a 64-bit Windows operation system with an i7-4790 CPU with 8 GB RAM. All coordinate values in this article are in millimeters. 

### 6.1. Simulation of Single Manipulator Avoidance

When $\mu_{A}=0$ or $\mu_{B}=0$, the dual manipulators avoidance strategy will be converted to a single manipulator avoidance strategy. Here, we define $\mu_{A}=0$ to show the simulation results of single manipulator avoidance. The space coordinates of the base of manipulator A are (0, 0, 0) while manipulator B’s base is situated at (0, −900, 0). Six different path planning cases and the corresponding simulation results are presented and compared in Figure 7 and Figure 8 intensively.

Case 1: the EEF of manipulator B stays at $P_{B}=\left(0,-450,\;500\right)$ while manipulator A moves from $P_{A}^{start}=\left(400,-450,\;500\right)$ towards $P_{A}^{end}=\left(-400,-450,\;500\right)$.

Case 2: the EEF of manipulator B stays at $P_{B}=\left(100,-450,\;500\right)$ while manipulator A moves from $P_{A}^{start}=\left(400,-450,\;600\right)$ towards $P_{A}^{end}=\left(-400,-450,\;400\right)$.

Case 3: the EEF of manipulator B stays at $P_{B}=\left(0,-450,\;500\right)$ while manipulator A moves from $P_{A}^{start}=\left(400,-400,\;500\right)$ towards $P_{A}^{end}=\left(-400,-500,\;500\right)$.

Case 4: the EEF of manipulator B stays at $P_{B}=\left(0,-450,\;500\right)$ while manipulator A moves from $P_{A}^{start}=\left(400,-450,\;400\right)$ towards $P_{A}^{end}=\left(-400,-450,\;600\right)$.

Case 5: the EEF of manipulator B stays at $P_{B}=\left(0,-500,\;500\right)$ while manipulator A moves from $P_{A}^{start}=\left(400,-450,\;500\right)$ towards $P_{A}^{end}=\left(-400,-450,\;500\right)$.

Case 6: the EEF of manipulator B stays at $P_{B}=\left(0,-400,\;500\right)$ while manipulator A moves from towards $P_{A}^{start}=\left(400,-500,\;500\right)$ towards $P_{A}^{end}=\left(-400,-500,\;500\right)$.

We can simulate the effect of the algorithm proposed in this paper on the path-planning strategy of a dual manipulator system with different initial paths by using the control variables method. Take Case 1 as a benchmark case, both Cases 2 and 4 add displacement in the *Z*-axis direction of the world coordinate system (WCS), distinguishing the two by the opposite direction of displacement. The displacement of the manipulator A in the *Z*-axis is a descent from the high to the low coordinate in Case 2, while in Case 4 the situation is reversed. Similarly, and in turn, Case 3 incorporates a *Y*-axis movement.

As the working space of an industrial manipulator is a finite three-dimensional space centered on the base of the manipulator, an avoidance strategy along the direction to the base is always the safest option for the obstacle avoidance of the manipulator. Figure 8 illustrates the path patterns of three different dual manipulator systems and the respective avoidance strategies. It is clear that Cases 1, 5, and 6 in Figure 7 are simulation avoidance results for each of these three path patterns.

The space coordinate vector of an industrial manipulator arm is usually comprised of two components $P_{B}=\left(\left[x,y,z\right],\left[\alpha ,\beta ,\gamma \right]\right)$, where [x,y,z] represents the 3D space coordinates of the centre point of the manipulator EEF and [α,β,γ] represents the pose direction vector of the manipulator EEF. The EEF pose vectors of the manipulator A and B are assigned the values $\left[\alpha_{A},\beta_{A},\gamma_{A}\right]=\left[\pi /2,-\pi /2,0\right]$ and $\left[\alpha_{B},\beta_{B},\gamma_{B}\right]=\left[-\pi /2,-\pi /2,0\right]$ for all cases in Figure 7 and Figure 8.

In order to verify the generality of our algorithm, we re-simulated all cases by simply changing the corresponding EEF pose vectors without changing their individual space coordinates. When the pose vector of the EEF of manipulator *A* and *B* are set to $\left[\alpha_{A},\beta_{A},\gamma_{A}\right]=\left[\pi /2,\pi ,0\right]$ and $\left[\alpha_{B},\beta_{B},\gamma_{B}\right]=\left[-\pi /2,\pi ,0\right]$, the corresponding simulation results are shown in Figure 9.

By comparing the simulation results for all cases in Figure 7, Figure 8 and Figure 9, some apparent and obvious consensus can be quickly reached. Despite different EEF pose vectors and different initial motion states, the algorithm proposed in this paper always can adaptively generate suitable and reasonable obstacle avoidance paths. However, due to changes in the EEF pose vectors, our algorithm also produces significantly different simulated planning paths for the same case. For example, in Cases 1 and 3, the simulation results in Figure 7 and Figure 8 both show apparently large ‘bends’ in the *Z*-axis of the WCS and relatively less pronounced ‘bends’ in the X-Y plane, whereas in Figure 9 the simulated paths have only small ‘fluctuations’ along the *Z*-axis and large bends in the X-Y plane. Furthermore, the simulated path results for Cases 2 and 4 in Figure 7 and Figure 9 are exactly opposite in the “bending” direction along the Z axis. This discrepancy is directly due to the fact that various pose vectors of the EEF directly affect the space occupation of the manipulator bodies or in particular the end links, thus leading our algorithm to adaptively plan different obstacle avoidance paths under the planning principle of shortest paths when facing different space occupancies.

Retrospectively, Figure 1 shows that the insights advanced in this paper are improved by the progressive sequence of the map search method → SbOSMS → SbUOSM. Thus, it is essential to undertake comparative simulations of these three algorithms. Figure 10 compares the simulation results of Case 1 in Figure 8 utilizing the three methods described above. Equivalently, Figure 11 and Figure 12 correspond to the comparative simulation results for Cases 5 and 6 in Figure 8, respectively.

By detailed comparison of Figure 10, Figure 11 and Figure 12, some obvious observational conclusions can be summarized briefly. In terms of the softness of the planned path curve, the proposal in this paper takes speed into account as a factor in the optimization and thus achieves a smoother solution than the map search method and SbOSMS, which only consider planning space path points. When it comes to the spatial complexity of planning path curves, the paths planned by the SbOSMS algorithm are only offset in the X-Y plane in the WCS in all three cases, the map search method (A*) is shifted in all axis directions in space, while the SbUOSM algorithm is only biased in the X-Y plane in case 1, and the remaining cases are similar to A*. This is a side effect of the greater flexibility with which our proposals operate in the face of different scenarios.

The total deviation of the planned path from the initial path is also a key indicator of the merit of the obstacle avoidance path-planning algorithm. Assuming that the equation of the planned path curve is $g_{0}=f_{0}\left(L\right)$ and the equation of the initial path curve is g=f(L), then the total offset of this planned path is:(11)$$\Delta_{g\to g_{0}}=\overset{L}{\underset{0}{\int }}∥f_{0}\left(t\right)-f\left(t\right)∥dt$$

All the planning results in Figure 10, Figure 11 and Figure 12 were used to calculate the total deviation utilizing the above equation and the results were assembled into Table 1.

Meanwhile, we simulate each of these three algorithms in the six cases mentioned above and assemble the respective algorithm execution time and the number of planning path steps into Table 2.

Comprehensive analysis of Table 1 and Table 2 clearly shows that the proposal in this paper is significantly better than the map search method and the SbOSMS algorithm in terms of the deviation of the planning path, the execution time of the algorithm and even the number of planning steps.

### 6.2. Simulation of Dual Manipulator Avoidance

Similar to single manipulator avoidance, 5 cases will be used to simulate the operation of the SbUOSM algorithm in a dual manipulator avoidance scenario.

Case 1: the EEF of manipulator A moves from $P_{A}^{start}=\left(400,-450,\;500\right)$ towards $P_{A}^{end}=\left(-400,-450,\;500\right)$ while manipulator A moves from $P_{B}^{start}=\left(-400,-500,\;500\right)$ towards $P_{B}^{end}=\left(400,-500,\;500\right)$.

Case 2: the EEF of manipulator A moves from $P_{A}^{start}=\left(400,-450,\;500\right)$ towards $P_{A}^{end}=\left(-400,-450,\;500\right)$ while manipulator A moves from $P_{B}^{start}=\left(-400,-450,\;500\right)$ towards $P_{B}^{end}=\left(400,-450,\;500\right)$.

Case 3: the EEF of manipulator A moves from $P_{A}^{start}=\left(400,-450,\;500\right)$ towards $P_{A}^{end}=\left(-400,-450,\;500\right)$ while manipulator A moves from $P_{B}^{start}=\left(-400,-400,\;500\right)$ towards $P_{B}^{end}=\left(400,-400,\;500\right)$.

Case 4: the EEF of manipulator A moves from $P_{A}^{start}=\left(400,-400,\;500\right)$ towards $P_{A}^{end}=\left(-400,-500,\;500\right)$ while manipulator A moves from $P_{B}^{start}=\left(-400,-400,\;500\right)$ towards $P_{B}^{end}=\left(400,-500,\;500\right)$.

Case 5: the EEF of manipulator A moves from $P_{A}^{start}=\left(400,-450,\;400\right)$ towards $P_{A}^{end}=\left(-400,-450,\;600\right)$ while manipulator A moves from $P_{B}^{start}=\left(-400,-450,\;400\right)$ towards $P_{B}^{end}=\left(400,-450,\;600\right)$.

Similarly, when we refer to Case 2 as the benchmark case, based on the fact that both manipulators have relative motions in the dual manipulator avoidance, Cases 2 and 4 in Figure 8 are fused into Case 5 in Figure 13, which is used to show how this paper’s proposal operates when adding the displacement of the initial path in the *Z*-axis direction of the WCS; In Figure 13, Cases 1–3 investigate the effect of the relative distance between two manipulators on the planning results of the algorithm, similar to cases 1, 5 and 6 in single manipulator avoidance; Case 4 intersects the paths of the two manipulators in the X-Y plane of the world coordinate system, similar to Case 3 in Figure 8, aiming to reveal the specific simulation results of the algorithm for displacement increments in the *Y*-axis direction in the WCS.

As the effect of changing the end-effector pose vector on the planning results has already been shown in the section on single-arm avoidance (Figure 9), this variable will not be modeled in this chapter.

Figure 14, Figure 15 and Figure 16 focus on the difference between the results of the three different algorithms. The analogous results for dual manipulator avoidance are similar to those for single manipulator avoidance. SbUOSM can always produce smoother paths. The results of the map search algorithm are always the most space-complex. Except for Figure 16, both SbUOSM and SbOSMS perform avoidance only in the *X*-*Y* plane of the world coordinate system. In Figure 16, the path points of SbUOMS are all dis-tributed in the X-Z plane, mainly because the dimensions of the end links are more distributed in the X-Y plane than in the X-Z plane.

All the planning results in Figure 14, Figure 15 and Figure 16 were used to calculate the total deviation utilizing the above equation, and the results were assembled in Table 3.

Meanwhile, we simulate each of these three algorithms in the six cases mentioned above and assemble the respective algorithm execution time and the number of planning path steps into Table 4.

Comprehensive analysis of Table 3 and Table 4 clearly shows that the proposal in this paper is significantly better than the map search method and the SbOSMS algorithm in terms of the deviation of the planning path, the algorithm’s execution time, and even the number of planning steps.

From Equation (1) we can also visualize that the planning results are also directly governed by the weighting factors $\mu_{A}$ and $\mu_{B}$. Figure 17 and Figure 18 illustrate visually the effect of different values of $\mu_{A}$ on the shape of the end-effector paths of manipulators A and B, respectively, in the form of multicolored graphs. In Figure 17 and Figure 18, the horizontal axis is the coordinate of the end of the corresponding manipulator under the X axis of the WCS, and the vertical axis is the coordinate of the end of the corresponding manipulator under the Y axis of the WCS. Due to the constraint $\mu_{A}+\mu_{B}=1$, when $\mu_{A}$ increases, $\mu_{B}$ decreases. Observing Figure 17 and Figure 18 together, as $\mu_{A}$ gradually increases from 0.1 to 0.9, the maximum offset of the path planned at the end of manipulator arm A on the *Y*-axis is incrementally reduced, and the opposite is true for manipulator B. From the geometric properties of the path curve, the curvature of the path of manipulator A enlarges and the curvature of the path of manipulator B diminishes.

## 7. Conclusions

In this paper, a novel path-planning method called SbOSMS was proposed, which can successfully facilitate the planning of non-collision paths for all manipulators working under dual manipulator cooperative manufacturing scenarios according to the real-time circumstance at any time while all manipulators are working along with their tasks without stopping. 

This paper inherits the main architecture of the SbOSMS algorithm, which also combines the map search method and the time-sampling method, and merely upgrades the local point map in a single sampling period to a global spherical map. To this end a system of multivariate vector inequality equations was constructed based on the spatial geometry information and motion properties of the dual manipulator system. However, after preliminary simplifications, the initial set of objective functions and constraint equations for vector inequalities are not directly convertible into a set of equations with the same variables. This paper converts all the equations into a system of equations with common vector variables by proposing a new minimum distance prediction method to linearly fit the minimum distance curve of a dual manipulator system. The three-dimensional properties of the variable vectors are then used to spatially geometrify the entire set of equations, using spatial geometry to reduce the entire set of vector programming equations to scalar non-linear optimization equations that can be easily solved. By comparing the simulation results of single manipulator avoidance and dual manipulator avoidance for various scenarios, the proposal in this paper can obviously be superior to SbUOSM and the earlier map search method from different perspectives such as the avoidance of path complexity degree, deviation, algorithm execution time, and the number of planning steps.

However, the paper has not probed further into how much the linear fitting treatment of the minimum distance curve for the dual manipulator system actually affects the entire planning result, which could be one of the future research points. At the same time, the application of this proposal to multi-manipulator systems is also a direction in which work needs to be invested.

## Figures and Tables

**Figure 1 sensors-22-02502-f001:**
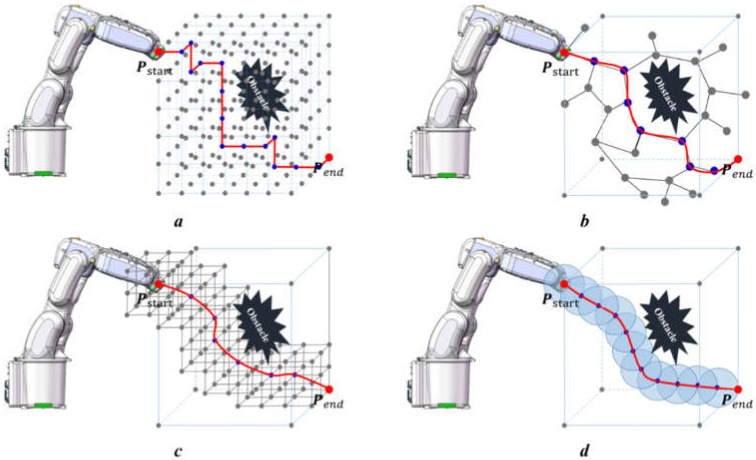
Schematic diagram of four different path-planning strategies for industrial manipulators. Picture (**a**) shows the planning strategy of map-based planning algorithms such as A*; Picture (**b**) shows the planning strategy of sampling-based planning algorithms such as RRT, PRM, and APF, etc.; Picture (**c**) is SbOSMS strategy which combines a map-based planning strategy and a sampling-based planning strategy to reduce both the search space and search time, as well as overcome the defect of random planning results for RRT; Picture (**d**) shows the proposal of this paper, which changes the node map in picture c into local-global sphere search space around EEF position point during every sampling period.

**Figure 2 sensors-22-02502-f002:**
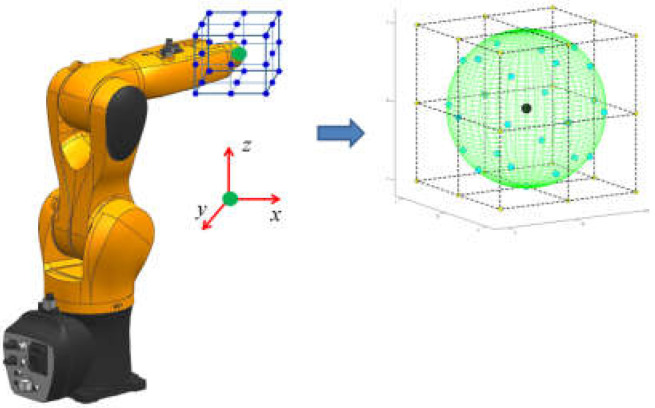
Schematic of a local search map within a sampling period in SbOSMS and our proposal. The set of all blue dots located on the surface of the green sphere in the right half of the diagram forms the search map for the SbOSMS method during a single sampling period. This paper attempts to extend the local search map within a single sampling period from these “special point sets” to the entire green sphere surface area.

**Figure 3 sensors-22-02502-f003:**
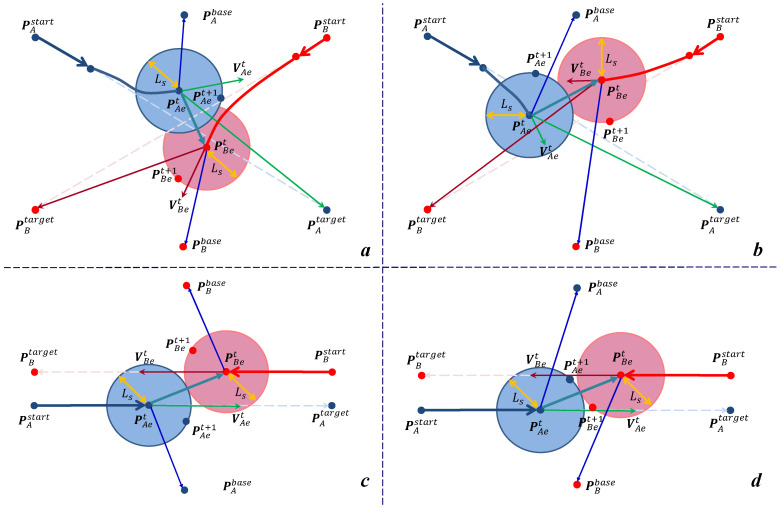
Influence of the relative positions of the dual manipulator EEFs on the obstacle avoidance path-planning strategy. The upper part of Figure 4 (picture **a**,**b**) of the dual manipulator system have crossed initial paths, while the bottom half (**c**,**d**) is set to initial parallel paths. Meanwhile, the left half of picture a and c corresponds to the interval scenario, (**b**,**d**) fall under the interact scenario specified in Figure 4. Different initial paths and coexistence scenarios give rise to various choices of obstacle avoidance points $P_{ie}^{t+1}\left(i\in \left\{A,B\right\}\right)$. For the interval scenario, the relative position vector ($P_{Be}^{t}-P_{Ae}^{t}$) between manipulator B and A is outside the working space of manipulator A (consisting of vectors ($P_{Ae}^{t}-P_{A}^{base}$), ($P_{A}^{target}-P_{A}^{base}$) and ($P_{A}^{target}-P_{Ae}^{t}$) together). In this case, it is only necessary to consider how to reach its terminal $P_{A}^{target}$ as quickly as possible while avoiding the obstacle to ensure that manipulator A has the safest tendency to move towards its base. However, when it comes to interact scenarios, the relative position vector is in the working space of manipulator A, then the first thing that should be considered for manipulator A is to evade immediately along the direction of its base and proceed to its target after it has wholly bypassed the danger area.

**Figure 4 sensors-22-02502-f004:**
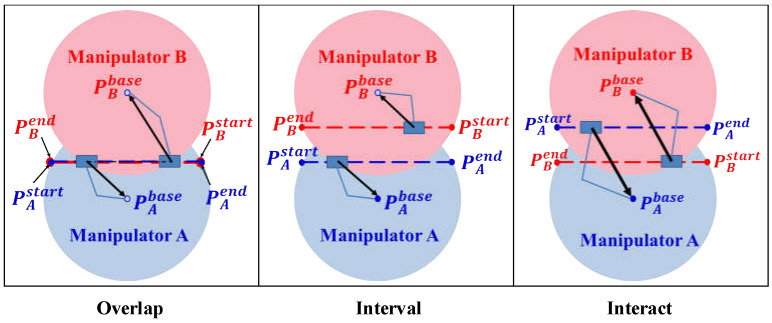
Three dual-manipulator coexistence scenarios. The blue dotted line represents the initial path of manipulator A, and the red dotted line represents the initial path of manipulator B. The blue area is the operation space of manipulator A, and the red area is the operation space of manipulator B. In the scenario of “Overlap”, the initial path of manipulator A is in the operation space of manipulator A and the initial path of manipulator B is in the operation space of manipulator B. In scenario “Interval”, the two initial paths overlap and the EEF points of the two manipulators should be driven back towards their points of departure. In scenario “Interact paths”, the avoidance strategy pushes the EEFs of the two manipulators into the working space of another manipulator, undoubtedly increasing the possibility of collision.

**Figure 5 sensors-22-02502-f005:**
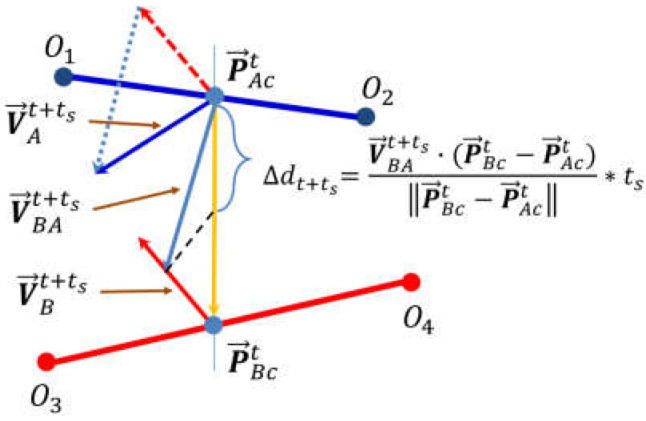
Schematic of minimum distance prediction for dual manipulators system within the sampling period ($t\sim t+t_{s}$). $O_{1}O_{2}$ and $O_{3}O_{4}$ denote the minimum distance links for manipulator A and B, and $P_{Ac}^{t}$ and $P_{Bc}^{t}$ are the closest points on the two minimum distance links, respectively. $V_{A}^{t+t_{s}}$ and $V_{B}^{t+t_{s}}$ are the displacement velocity vectors of $P_{Ac}^{t}$ and $P_{Bc}^{t}$ at the current moment t. Here, we simplify the non-linear movement of the minimum distance point along the link to a linear advection, so that $\Delta d_{t+t_{s}}$ can be easily approximated using dynamic spatial geometry.

**Figure 6 sensors-22-02502-f006:**
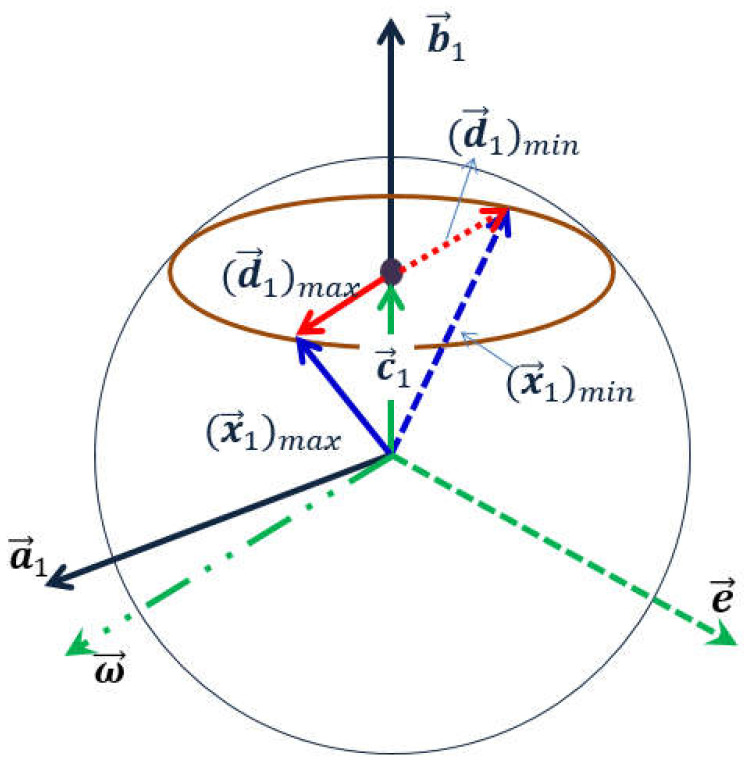
The diagram of vector conversion relation of the optimal Equation (9).

**Figure 7 sensors-22-02502-f007:**
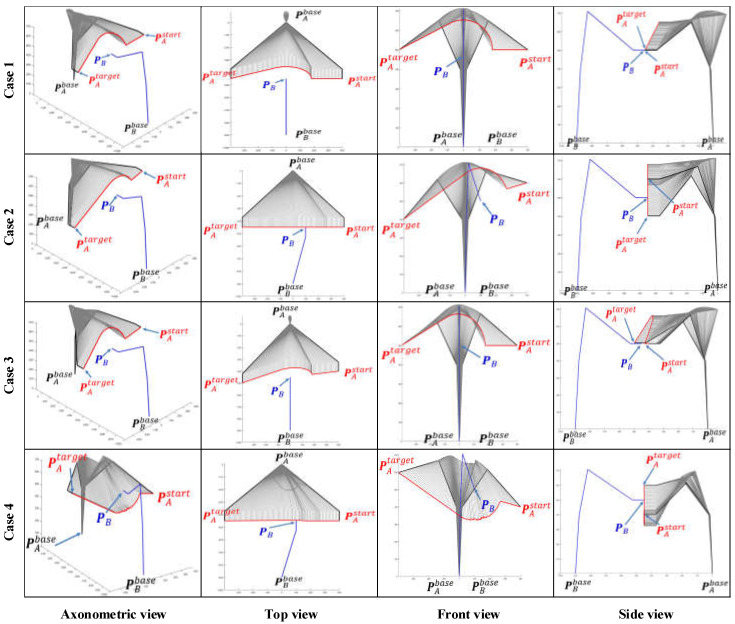
Comparison of single manipulator avoidance simulation results for four initial path cases. As the simulation results are 3D path curves, the results are presented in a comprehensive and extensive visualization with four views: axonometric, top, front, and side views.

**Figure 8 sensors-22-02502-f008:**
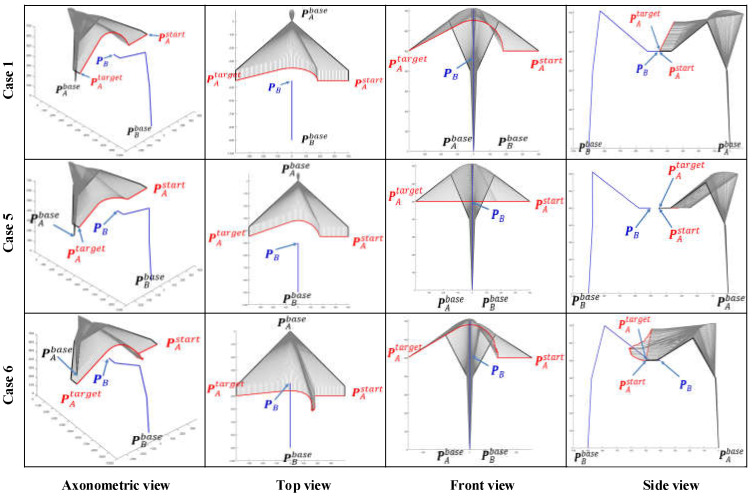
Comparison of single manipulator avoidance simulation results for three initial path cases. These three cases yield three completely different simulation results by changing the coordinates of manipulator A on the *Y*-axis of the WCS while maintaining the same spatial position of manipulator B.

**Figure 9 sensors-22-02502-f009:**
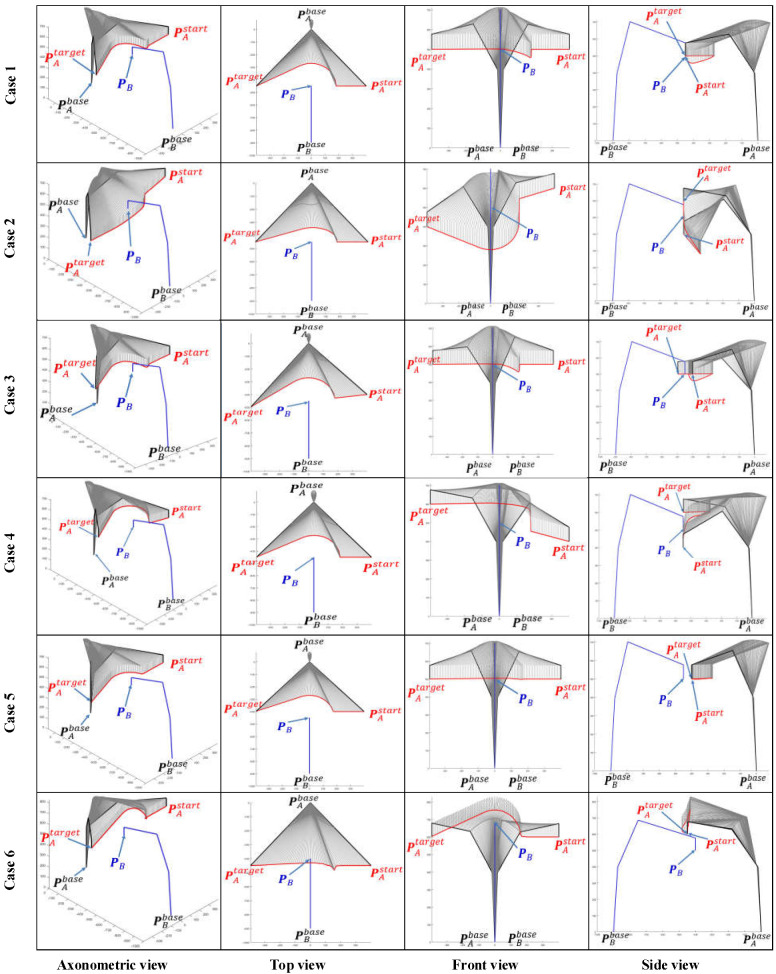
Comparison of single manipulator avoidance simulation results. Compared to Figure 7 and Figure 8, the six cases in this diagram merely modify the end-effector pose vectors of the manipulators, whilst the simulation results obtained are significantly distinctive.

**Figure 10 sensors-22-02502-f010:**
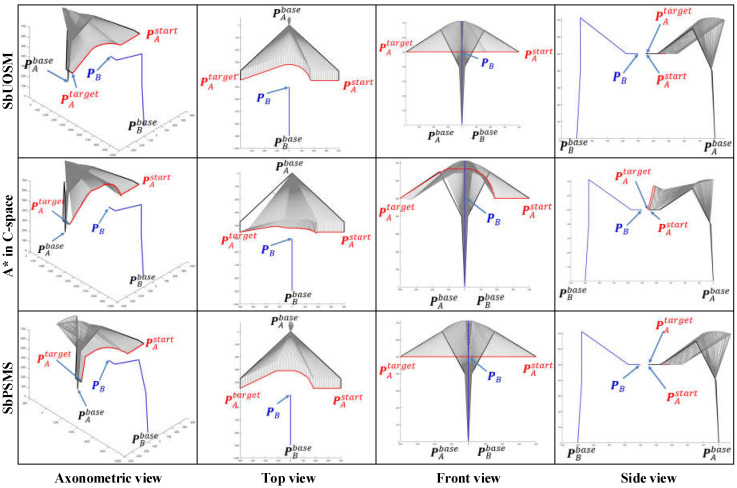
Comparison of simulation results of three path-planning algorithms for Case 1.

**Figure 11 sensors-22-02502-f011:**
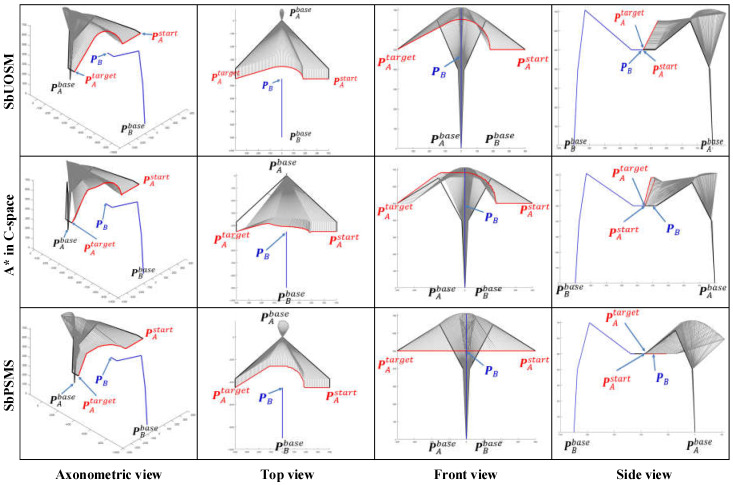
Comparison of simulation results of three path-planning algorithms for Case 5.

**Figure 12 sensors-22-02502-f012:**
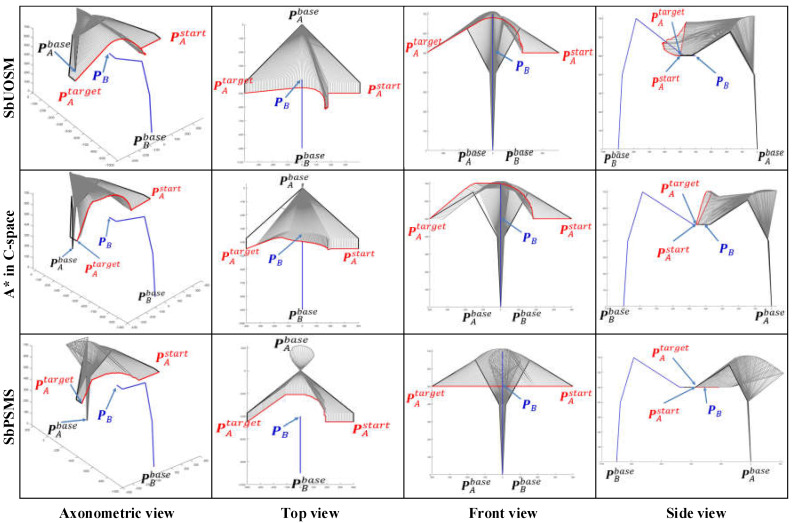
Comparison of simulation results of three path-planning algorithms for Case 6.

**Figure 13 sensors-22-02502-f013:**
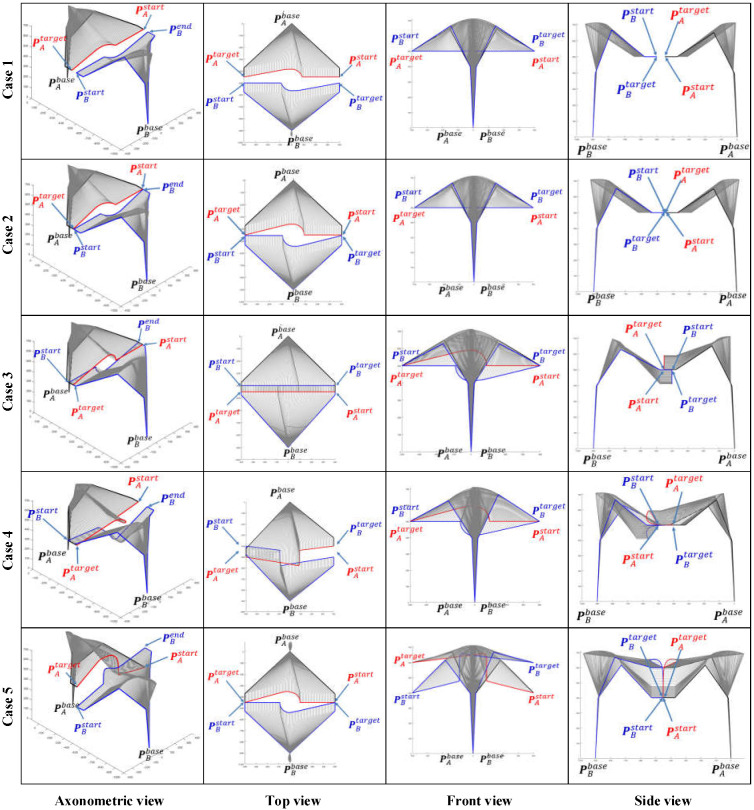
Comparison of dual manipulators avoidance simulation results for five initial path cases. As the simulation results are 3D path curves, the results are presented in a comprehensive and extensive visualization with four views: axonometric, top, front and side views.

**Figure 14 sensors-22-02502-f014:**
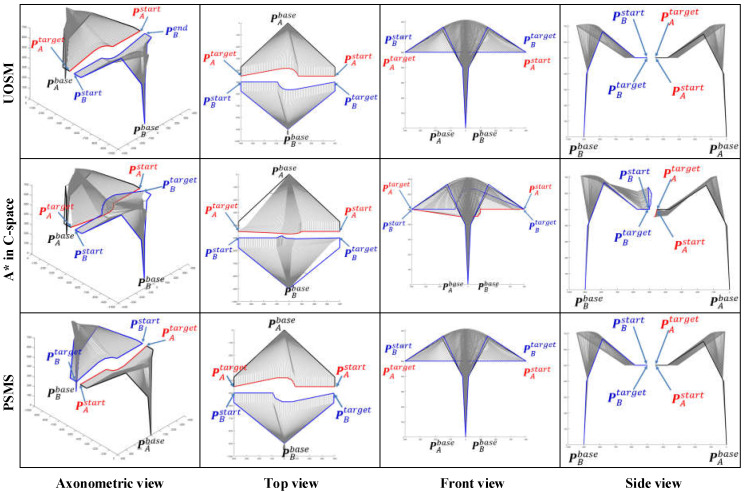
Comparison of simulation results of three path-planning algorithms for Case 1.

**Figure 15 sensors-22-02502-f015:**
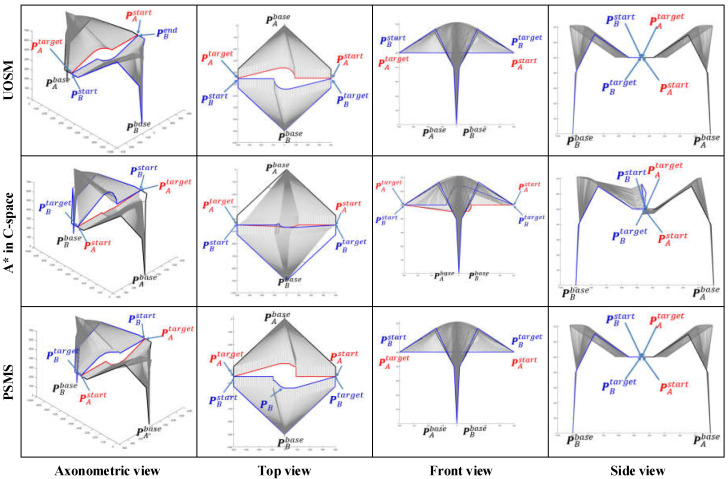
Comparison of simulation results of three path-planning algorithms for Case 2.

**Figure 16 sensors-22-02502-f016:**
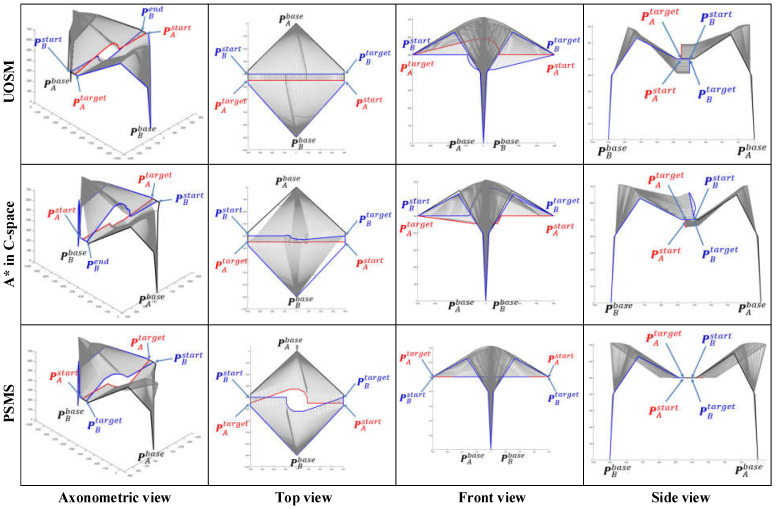
Comparison of simulation results of three path-planning algorithms for Case 3.

**Figure 17 sensors-22-02502-f017:**
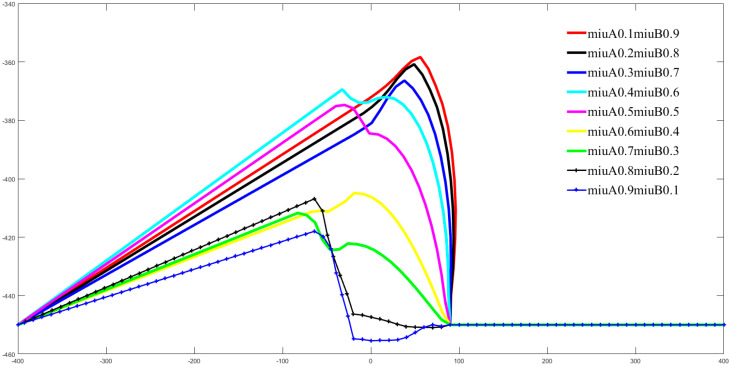
For Case 1 in Figure 13, the paths of the manipulator A EEF are simulated for different values of the positive circumventing weighting factors $\mu_{A}$ and $\mu_{B}$.

**Figure 18 sensors-22-02502-f018:**
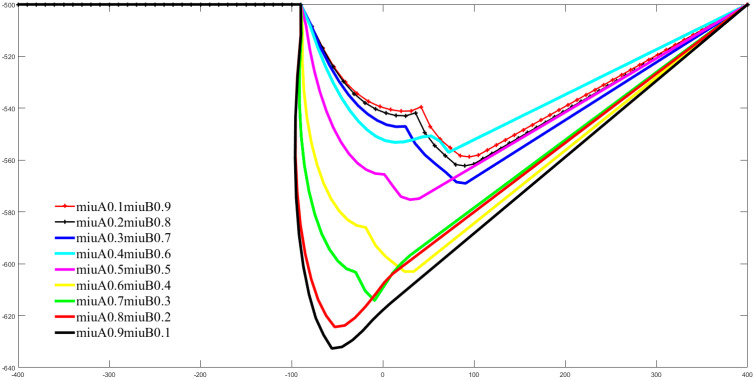
For Case 1 in Figure 13, the paths of the manipulator B EEF are simulated for different values of the positive circumventing weighting factors $\mu_{A}$ and $\mu_{B}$.

**Table 1 sensors-22-02502-t001:** Summary table of planning path deviations for the three algorithms under all six cases (Unit: mm).

	Case 1	Case 2	Case 3	Case 4	Case 5	Case 6
SbUOSM	4899.4	7183.7	7619.6	8185.9	7586.8	10,844
A* in C-space	13,161	20,914	19,047	21,187	16,150	19,153
SbOSMS	5814.5	8461.6	9015.4	8794.1	8629.2	11,878

**Table 2 sensors-22-02502-t002:** Summary table of algorithm execution times/total planning steps for the three algorithms under the six cases (Unit: μs/step).

	Case 1	Case 2	Case 3	Case 4	Case 5	Case 6
SbUOSM	6876.7/95	6825.9/93	7703.3/95	7045.5/96	6405.9/88	8892.3/121
A* in C-space	45,992.6/145	38,291.4/131	48,498.9/151	38,153.8/131	42,050.7/138	46,764.1/142
SbOSMS	13,477/95	13,313.9/93	13,148.3/97	13,340.8/97	12,391/89	17,043.8/112

**Table 3 sensors-22-02502-t003:** Summary table of planning path deviations for the three algorithms under all six cases (Unit: mm).

	Case 1	Case 2	Case 3	Case 4	Case 5
SbUOSM	3944.9	5457.9	5647.5	4857.7	5846.8
A* in C-space	8341.2	9276.3	9373.5	8887.8	9006.1
SbOSMS	4280.3	6162.2	8126.8	5964.7	6765.3

**Table 4 sensors-22-02502-t004:** Summary table of algorithm execution times/total planning steps for the three algorithms under the six cases (Unit: μs/step).

	Case 1	Case 2	Case 3	Case 4	Case 5
SbUOSM	16,650.6/90	16,029.7/88	16,344.1/88	18,334.9/100	18,641.7/103
A* in C-space	68,246.3/107	67,822.4/108	65,255.6/106	69,112.5/107	67,594.8/104
SbOSMS	50,392.1/85	58,154.9/91	66,888.6/92	54,660.6/91	57,083.3/92

## Data Availability

The authors confirm that the data and materials supporting the findings of this study are available within the article.

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
