# Peer review of "A Sampling-Based Unfixed Orientation Search Method for Dual Manipulator Cooperative Manufacturing"

_sensors, 2022, doi:10.3390/s22072502_

Round 1

Reviewer 1 Report

This paper investigate A Sampling-based Unfixed Orientation Search Method for Dual manipulator cooperative manufacturing. This is a well-written and well-researched article. It has a good theoretical support and simulation verification part. There are a few minor issues that I feel need to be corrected before publication.

  1. The abstract section, as well as the intro section, should more clearly state the innovative points of this paper.
  2. Please note the issue of the resolution of the image, in the version I can see, Figure 4 is rather blurry.
  3. Authors need to take further care to check language issues and avoid using some long sentences.
  4. Check the formatting of formulas, e.g. from 217-222, 333-337, etc.

Reviewer 2 Report

The paper is publishable in principle. However, I have a couple of comments that will hopefully be appreciated to improve the impact of this article.

The abstract could highlight the contribution to knowledge better. The paper highlights benefits such as being smoother and having less deviations (meaning higher accuracy?). However, these are incremental benefits of an already existing algorithms. So, what is the research question this article is addressing that could make a breakthrough or even agenda setting difference? 

The figure 1 is very helpful. However, in the back e.g. on page 13, 15 ff. there are a lot of figures that are hard to read and quite repetitive. I suggest maybe only showing those figures that contribute to knowledge most. Figure 3 uses too small font. 

I wondered whether the radii are determined also considering the speed of the robotic arms. A smoother movement of the arms may help with reducing joint wear?

As this project is sponsored by an industry partner, I would assume that it would be possible to practically test the algorithm using real robots. The simulations/calculations could be supplemented with some targeted real-world validations or other means of linking theory with reality to increase the impact of this article. 

Reference on page 8 line 267 missing. 

Some formatting bugs where a title is at the bottom of a page (e.g. page 9). 

Reviewer 3 Report

The Authors propose a sampling-based unfixed orientation search method for dual manipulator cooperative manufacturing.  The manuscript is well written and organized. The literature review presented in the introduction has been properly conducted with respect to the motivation of the problem and the techniques that have been used. The method is clearly described and the simulation results are appropriately analyzed.

Please, take into account the following minor comments and suggestions.

Please, provide more quantitative results that support the fact that the proposed algorithm can be used in real-time.

A reference is missing in line 266.

There are several typos; e.g. lines 379 and 380.

Please, correct 1495 on line 19.

In section 3.2 provide the units of the coordinates for each of the proposed cases.

To increase the impact of the manuscript, I recommend including an example of a practical application of the proposed approach.

Reviewer 4 Report

The study presents an up-to-date and relevant research topic, which is absolutely fit to the profile of the journal.

The structure of the study is logical and clearly presented:

- The Abstract contains the description of the purpose and results of the research.

- The Introduction briefly places the study in a broad context and highlight why the research is important. The publications, which are cited in the paper support to understand why this research topic is relevant and provide a good overview of the literature. The purpose of the research is clearly defined relating to the research tasks.

- In the main parts of the article (3. Mathematic model; 4. Minimum distance prediction; 5. Sampling-Based Unfixed Orientation Search Method; 6. Simulation and analysis) a novel path planning method called “SbOSMS” is proposed, which combines the map search method and the time-sampling method. In my opinion the new algorithm is valuable, and described in detailed. The efficiency of the method is confirmed by a comparative study.

- Section „Conclusions” introduces the evaluation and main findings of the research.

In my opinion the article needs to be improved before publication. My comments and suggestions are the following:

1.)  The numbers of the Sub-sections of Section 1 and Section 6 are not correct. Please modify these.

2.) The quality of the Figures 7-16. can be improved.

3.) Figures 17-18. can be evaluated in more detailed; furthermore, the dimension (unit) is missing.

4.) A “List of symbols” Section could help the readers to understand easily the equations. This list can be inserted e.g. into Appendix.

5.) The efficiency improvement of the new method can be concluded also in time saving (in “time” unit).

6.) Please use up-to-date literature in the reference list. The most of the references are old.

7.) Please apply the requirements of the journal’s template in case of the style of the reference list.

Based on the before mentioned comments I suggest the acceptance of the paper after minor revision.
